# Ionic Liquid-Microwave-Based Extraction of Biflavonoids from *Selaginella sinensis*

**DOI:** 10.3390/molecules24132507

**Published:** 2019-07-09

**Authors:** Dan Li, Chengxin Sun, Jiaqiang Yang, Xiankui Ma, Yongmei Jiang, Shaoliang Qiu, Gang Wang

**Affiliations:** School of Pharmacy, Zunyi Medical University, Zunyi, Guizhou 563003, China

**Keywords:** *Selaginella sinensis*, ionic liquids, amentoflavone, hinokiflavone, microwave extraction, HPLC, response surface methodology

## Abstract

*Selaginella sinensis* (Desv.) Spring has been used for many years as traditional Chinese medicine (TCM) for many years. Recently, ionic liquids (ILs) have attracted great attentions in extraction and separation technology of TCM as a new green solvent. In this paper, microwave assisted extraction-IL (MAE-IL) that extracted amentoflavone (AME) and hinokiflavone (HIN) from *Selaginella sinensis* was reported for the first time. The contents of two biflavonoids were simultaneously determined by a high performance liquid chromatography (HPLC) method. After different ionic liquids were compared, it was found [C_6_mim]BF_4_ had a high selectivity and efficiency. Moreover, the important extraction conditions, including solid-liquid ratio, IL concentration, extraction time, microwave power and radiation temperature, were also investigated and optimized by response surface methodology (RSM) using AME and HIN yields as index. The results showed that the extraction yields of AME and HIN from *S. sinensis* were 1.96 mg/g and 0.79 mg/g, respectively, under the optimal process parameters (0.55 mmol/L, 300 W, 40 min, 1:11 g/mL and 48 °C). Compared with the conventional extraction methods, MAE-IL could not only achieve higher yield in shorter time, but also could reduce the consumption of solvent. This effective, rapid and green MAE-IL method was suitable for the extraction of AME and HIN.

## 1. Introduction

*Selaginella sinensis* (*S. sinensis*), the dried whole grass of *Selaginella sinensis* (Desv.) Spring, grows in Northeast China and belongs to selaginell genus [1]. It is mainly used to treat coughs, laryngopharyngeal swelling, rheumatism, prostatitis, chemical cystitis, traumatic bleeding and dysentery [2]. In the reported articles, different components have been found from *S. sinensis*, including biflavonoids, alkaloids, sterols and organic acids [3]. Biflavonoids is the major chemical constituents of *S. sinensis*, which is consisted with two mono flavonoids with complex and diverse structural characteristics [4]. Among them, amentoflavone (AME) and hinokiflavone (HIN) (as shown in Figure 1) are the active ingredients in *S. sinensis* [5] and have a wide range of pharmacologic activity, such as anticancer, antiviral, anti-inflammatory, anti-Alzheimer's disease, anti-myocardial ischemia and antimicrobial effects [6,7]. 

AME has remarkable anti-inflammatory properties that have effect on all kinds of proinflammatory cytokines. For instance, AME could inhibit the upregulation of prostaglandin E2 in psoriatic via down-regulation of cyclooxygenase-2 (COX-2). Lee and colleagues [8] found that AME, podocarpusflavone A and 4′, 7-dimethyl AME had obvious inhibitory effects on cathepsin B. Their IC_50_ values were 1.75, 1.68 and 0.55 μmol/L, respectively, which were known as new natural inhibitors of cathepsin B. The study of Heo and others [9] found AME from *Selaginella tamariscina* had remarkable inhibition effect against respiratory syncytial virus (RSV) pneumonia (IC_50_ = 5.5 μg/mL), and herpes simplex virus (HSV) (IC_50_ = 6.5 μg/mL). Yang and others [10] proved HIN could inhibit cell proliferation of human melanoma A375 and CHL-1 cells in a dose- and time-dependent manner. Furthermore HIN could obviously enhance reactive oxygen species (ROS) and decrease the mitochondrial membrane potential. Coulerie P [11] and others’s study firstly revealed that biflavonoids from the leaf extract of *Dacrydium balansae* could strongly inhibit the dengue 2 virus RNA-dependant RNA polymerase and HIN had the strongest inhibiting effect among four biflavonoids.

Ionic liquids (ILs) is an organic molten salt under low temperature, which is composed entire of structural asymmetric anions and cations [12]. Compared with traditional organic solvents, ILs have the advantages of lower steam pressure, wider viscosity range, better electrical conductivity, stronger solubility and higher thermal stability [13], which has become popular as a environmental-friendly solvent. Recently, more and more ILs have been used in the extraction and analysis of traditional Chinese medicine (TCM). The interactions between ILs and chemical components of TCM mainly includes hydrogen bonding, hydrophobic effect, electrostatic action and dipole-dipole interaction [14].

Microwave-assisted extraction (MAE) combined with ILs can more effectively extract organic active molecules from TCM [15], which may be that ILs as a solvent can well absorb and transfer microwave energy. MAE-ILs has many advantages including short extraction time, high extraction efficiency, low energy consumption and good recyclability [16]. And some scholars have noticed that various combinations of different anions and cations of ILs can affect the extraction efficiency of natural medicine [17]. For instance, according to the report of Li et al. [18], MAE-IL can not only improve the extraction efficiency of total biflavonoids in shorter time but also use less organic solvents compared to MAE and Soxhlet extraction with conventional organic solvents, which illustrated that IL has the potential to replace the latter.

Based on the above background, the aim of this study is to develop a convenient, effective and rapid MAE-IL method to extract AME and HIN from *S. sinensis*. After nine different ILs would be investigated and compared in this paper, the related effects of ILs structure and some important extraction conditions would be explored on the extraction efficiency for target compounds. Furthermore, the extraction parameters were optimized by response surface method (RSM). Finally, the optimized MAE-IL method was compared with several conventional extraction ways.

## 2. Results and Discussion

### 2.1. Screening Different ILs

The screening of IL’s anions and cations for high extraction efficiency of active ingredients from natural products are considered to be crucial for their various properties and performance [19]. Firstly, the 1-butyl-3-methylimidazolium ILs with five different anions (Br^−^, BF_4_^−^, OAC^−^, Cl^−^, NO_3_^−^) were chosen to study the influence of extraction yields of two biflavonoids. As shown in (Figure 2A), The yields of AME and HIN using IL with BF_4_^−^ were higher than those of other ILs with the same cation, which might be that the IL with BF_4_^−^ can quickly permeate into the cells and effectually dissolve target products by H-bond generated [20]. And then different 1-alkyl-3- methylimidazolium cations (C_2_mim^+^, C_4_mim^+^, C_6_mim^+^, C_8_mim^+^ and C_10_mim^+^) with the same of anion of BF_4_^−^ were compared for the yields of AME and HIN. Figure 2B illustrated that [C_6_mim]BF_4_ had remarkable extraction efficiency towards the two targets among these five ILs. Considering the improvement of hydrophobic effect [21], the content of AME and HIN were obviously enhanced with increasing alkyl chain lengths. But when carbon chain lengths exceeded C-8, AME and HIN yields were clearly decreased in that the enhancement of stereo hindrance effect [22]. Based on research above, [C_6_mim] BF_4_ was chosen for extraction of AME and HIN from *S. sinensis*.

### 2.2. Single Factor Experiment

Based on the single factor experiments, [C_6_mim] BF_4_ was used as the extractant. Furthemore, the probable influence parameters, including IL concentration, solid-liquid ratio, irradiation time, microwave power and extraction temperature, were evaluated in detail. According to the literature of He et al. [23], imidazolium ILs have the highest solubility in a series of ethyl alcohols at the same temperature, so ethanol was chosen to dissolve ILs.

#### 2.2.1. Effect of IL Concentration

IL concentration can be regarded as the primary influence factor for effective constituents of natural products [24]. The combination of IL and ethanol can rapidly destroy the plant cell membrane and dissolve a large number of target components in the cell, which can improve the extraction rate of target components [25]. In the following investigation, 80 mL of [C_6_mim] BF_4_ solutions with 0.2~1.2 mmol/L were compared during the extraction of 8.0 g *S. sinensis* powders at 300 W for 40 min. As shown in Figure 3a, the extraction yields of AME and HIN was linearly proportional to the concentration of (C_6_mim)BF_4_ in the range of 0~0.6 mmol/L, which might be that [C_6_mim] BF_4_ solutions can easily break the structure of cell membranes, dissolve more AME and HIN and promote the extraction of active gredients. But in the range of 0.6~1.2 mmol/L, the extraction efficiency of AME and HIN was still constantly decreased as a result of the viscosity increase of IL solution [26]. Thus, [C_6_mim] BF_4_ concentration of 0.2, 0.4 and 0.6 mmol/L were evaluated for the subsequent RSM optimization.

#### 2.2.2. Effect of Extracting Temperature

Extraction temperature is also a great influence factor on targets extraction of natural products. For natural active gredients, the increase of temperature can promote their molecular motion, enhance their solubility further and improve the extraction efficiency [27]. Here 80 mL of 0.8 mmol/L [C_6_mim] BF_4_ ethanol solution was employed in the extraction of 8.0 g *S. sinensis* powders at 300 W for 40 min, and the suitable extraction temperature was investigated from 20 to 70 °C in the following investigation. And then it could be found from Figure 3b that, in the range of 20~50 °C, the extraction efficiency of AME and HIN would continue to rise with the increase of temperature as a result of the increased solubility and permeability of [C_6_mim] BF_4_ solution. However, in the range of 50 to 70 °C, the extraction performance of AME and HIN was distinctly reduced, which might be that the two biflavonoids contained thermal-unstable phenolic hydroxyl, and could be oxidized at high temperature [28]. The result indicated that the extracting temperature of 40, 50, and 60 °C would be investigated in the next experiment of RSM.

#### 2.2.3. Effect of Solid-Liquid Ratio

Ratio of solid (herbal powders) to liquid (extractant) is key for the extraction process of Chinese herbal medicine, which changes the contact area between solids and liquids, dissolves a lot of the active ingredients and enhances the extraction efficiency by permeating into cells [29]. Here 0.8 mmol/L [C_6_mim] BF_4_ ethanol solution was used in the extraction of 8.0 g *S. sinensis* powders at 50 °C and 300 W for 40 min, and the suitable solid-liquid ratio was evaluated from 1:5 to 1:15 (g/mL) in the subsequent experiment. As shown in Figure 3c, in the range of 1:5–1:11, the extraction yields of AME and HIN would continue to add with the improvement of solid-liquid ratio. Whereas in the range of 1:11–1:15, the extraction performance was constantly decreased with the increase of solid-liquid ratio, which could be the result of producing a lot of impurities [30]. Therefore, the result indicated that the solid-liquid ratio of 1:9, 1:11 and 1:13 g/mL were suitable for further evaluated for complete extraction of AME and HIN from *S. sinensis*.

#### 2.2.4. Effect of MAE Power

After screening the effects of IL concentration, radiation temperature and solid-liquid ratio, the influence of microwave power on extraction yields of AME and HIN was also studied. In short, low microwave power could result in the incomplete extraction for AME and HIN, but excessive radiation power would be unnecessary and increased energy consumption after reaching complete extraction [31]. Hence, it was necessary to search a suitable level on the basis of the above screening results. As shown in Figure 3d, the full extraction could be achieved at 300 W and the maximum yields of AME and HIN were 1.90 mg/g and 0.73 mg/g, respectively. 

#### 2.2.5. Effect of Extraction Time

Compared with some traditional methods such as soxhelt extraction and percolation extraction, it is well-known that microwave assisted extraction can shorten radiation time and improve extraction efficiency. In general, if extraction time is too short, it will result in incomplete extraction of target composition; on the contrary, excess time will lead to the unnecessary energy consumption [32]. Thus in the following investigation, 0.8 mmol/L of [C_6_mim] BF_4_ solutions (80 mL) within 10~60 min were chosen and evaluated using the extraction of 8.0 g *S. sinensis* powders at 50 °C and 300 W. Figure 3e indicated that AME and HIN yields were obviously increased from 10 min to 40 min, then AME and HIN content in the extract had almost no change with further increase of extraction time. 

### 2.3. Analysis of Response Surfaces

RSM is a multivariate nonlinear regression method, which is widely applied in the optimization of natural products extraction [33]. After investigation of single factor experiment, IL concentration (0.4, 0.6, 0.8 mmol/L), solid-liquid ratio (1:9, 1:11, 1:13 g/mL) and radiation temperature (40, 50, 60 °C) were chosen as main influential parameters and further optimized via Box-Behnken design of RSM using AME and HIN yields as index. Quadratic polynomial fitting models using design-expert 10.0 software for the yields of AME (Y_1_) and HIN (Y_2_) were presented as the following equation:Y_1_ = 1.91 − 0.19X_1_ − 0.073X_2_ − 0.040X_3_ − 0.050X_1_X_2_ − 0.030X_1_X_3_ + 0.000X_2_X_3_ − 0.25X_1_^2^ − 0.21X_2_^2^ − 0.16X_3_^2^(1)
Y_2_ = 0.74 − 0.085X_1_ − 0.040X_2_ − 0.023X_3_ − 0.020X_1_X_2_ − 0.015 X_1_X_3_ − 5.0 × 10^-3^ X_2_X_3_ − 0.14 X_1_^2^ − 0.13 X_2_^2^ − 0.055 X_3_^2^(2)
where X_1_, X_2_ and X_3_ represented extraction time, solid-liquid ratio and IL concentration, respectively. ANOVA datas of quadratic polynomials were shown in Table 1 and Table 2. The results indicated that the two model were remarkable significance with *p* < 0.0001 and the lack of fit for two biflavonoids were not significant with *p* > 0.05. This model for AME and HIN indicated a high degree of correlation between the observed and predicted values with *R*^2^ = 0.9557 and 0.9656, respectively (See Figure 4). X_1_, X_2_, X_3_ had the greatest influence on the extraction of AME and HIN according to their *p* values <0.01. Therefore, the models were suitable to predict accurately the yields of the two biflavonoids.

The 3D models of RSM for target products could imply how two dependent variables affect the test results. Experiments were performed to show that, when one factor always remains on a set level, the interactive influence between other two variates revealed well from 3D RSM plots. As shown in Figure 5, the ordinate exhibited AME or HIN yields and the abscissa exhibited three variates (IL concentration, solid-liquid ratio, radiation temperature). The results showed that all RSM figures were upward convex with a maximum value of AME or HIN yields, which displayed the rationality of the prediction 3D models [34]. According to the quadratic model equation of the two biflavonoids, the optimal process parameters for AME and HIN were as follows: 0.55 mmol/L IL concentration, 1:11.15 g/mL solid-liquid ratio and 47.87 °C ultrasonic temperature, giving a predicted AME and HIN yield of 1.9512 mg/g and 0.7759 mg/g, respectively. Considering synthetically the optimum craft parameters in regard to AME and HIN content, the calibration tests were suitably modified at the optimal conditions of 0.55 mmol/L IL concentration, 1:11 g/mL ratio of solid to liquid and 48 °C ultrasonic temperature. The optimum yields of AME and HIN were 1.96 mg/g and 0.79 mg/g, respectively, which was basically the same with the predicted value and indicated that the model was used for predicting the optimized extraction process of AME and HIN from *S. sinensis*.

### 2.4. Recovery of Two Biflavonoids and IL

Various insoluble organic solvents can be considered as the ideal reagents to recover the target and IL [35]. After optimizing the parameters of MAE-IL, the insoluble organic solvents were screened for the extraction of AME and HIN from IL extracts. The recovery of two biflavonoids and IL was calculated respectively based on the concentration ratio of different target compounds before and after extraction by HPLC analysis data (as shown in Figure 6). According to the calculation results (See Table 3), ethyl acetate could acquire higher recovery of two biflavonoids and IL, which indicated that the vast majority of AME (92.17 ± 0.38%), HIN (91.53 ± 0.49%) and IL (95.28 ± 0.37%) were recovered from *S. sinensis* extract.

The performance of recovered [C_6_mim]BF_4_ in subsequent extractions was investigated in detail for the extraction of AME and HIN under the optimal conditions. As shown in Figure 7, it was found that the extraction yields of AME and HIN showed a slight decrease with the increase of reuse cycles. The results indicated that [C_6_mim]BF_4_ could be recycled and applied for the extraction of these two biflavonoids more than six times because around 90% of the satisfactory extraction yields were obtained in the sixth cycle. Thus, IL solution is proved to be a ideal solvent in MAE of the biflavonoids from *S. sinensis*, which has apparent advantages of environmental-friendness, solvent saving, low energy consumption and also good recyclability.

### 2.5. Comparison of Four Extraction Methods

The established MAE-IL process was compared with the conventional MAE, SE and PE approach in order to evaluate the extraction performance of AME and HIN using ILs. Before different extraction approaches were compared, different extractants (ethanol, ethanol-H_2_O (1:1, v/v), methanol, methanol-H_2_O (1:1, v/v) and acetone-H_2_O (1:1, v/v)) were investigated in detail with several conventional methods. Among them, Figure 8 showed ethanol could obtain high yields of AME and HIN with MAE, SE and PE approach and was suitable for further evaluated for the comparison between MAE-IL and the conventional methods.

The best extraction conditions were adopted for several methods as described earlier. In short, experiment conditions of three methods were as follows: MAE (extraction time 40 min and solvent to solid ratio 11:1 mL/g), soxhlet extraction (SE) (extraction time 120 min and solvent to solid ratio 15:1 mL/g), and percolation extraction (PE) (extraction time 12 h and solvent to solid ratio 20:1 mL/g).

As shown in Figure 9, compared with SE and PE approach, MAE-IL not only increased AME and HIN content by 2 to 5 times but also shortened the extraction time to 1/3 and saved solvent to 1/2. Moreover, extraction time and extractant consumption of two methods were basically the same, AME and HIN yields of MAE-IL were increased nearly two times compared to conventional MAE method. Through the comparison of extraction effect for four methods, it could be concluded that MAE-IL had a good potential value to be a quick, efficient and environment friendly extract method for AME and HIN content from *S. sinensis*.

### 2.6. Feedstock Analysis Before and After Extraction

In order to investigate the morphological characterizations of the herbs before and after extraction in the studied process, scanning electron microscope (SEM, Hitachi TM 3000, Tokyo, Japan) was successively applied to observe the characterize of *S. sinensis* and its residue after extraction by MAE-IL. As shown in Figure 10, the untreated raw materials (Figure 10A1,A2) obviously had a relatively smooth surface; thick cell wall and distinct boundary could be found between different tissues. However, after MAE-IL, *S. sinensis* powders by SEM exhibited a rough and porous cell wall structures and indistinct boundary between different tissues (Figure 10B1,B2). It indicates that microwave produced a lot of radiation at high temperature and destroyed greatly the cell structure of plant material, which could make IL solution penetrate rapidly into the cell wall and increase diffusion of target products into the solvent. Thus, MAE-IL is a very effective method for permeation and diffusion of targets, and the improved mass transfer is closely related with high AME and HIN yields from *S. sinensis*.

### 2.7. Method Validation

In order to ensure reliable and precise analysis results of MAE-IL extraction, methodological investigation of HPLC, including linearity, stability, precision, repeatability and recovery, must be established using AME and HIN yields as index under the optimized conditions.

After a series of standard solutions of AME and HIN with different concentration levels were prepared, the linearity was firstly assessed for two biflavonoids and the calibration curves were established by HPLC. The standard curves of AME and HIN were Y_AME_ = 346.35X + 3.4667 (*R*^2^ = 0.9992) and Y_HIN_ = 264.05X + 6.0667 (*R*^2^ = 0.9996), respectively, where X was their concentration (μg/mL) and Y was the value of peak area. As shown in Table 4 the results showed well linearity with correlation coefficients (*R*^2^) more than 0.999, and LOQs (S/N = 10) and LODs (S/N = 3) for AME and HIN of the extracts were less than 0.278 μg/mL and 0.046 μg/mL, respectively. 

Stability, intraday precision, interday precision, repeatability and recovery of the optimal extraction condition were confirmed for each *S. sinensis* extract. The stability was assessed on the basis of determination of each analyte at 0, 12, 24 and 36 h at 25 °C, and the relative standard deviation (RSD) of peak area were 0.68% and 1.21% for AME and HIN, respectively, which indicated that the two biflavonoids were stable in IL extract at 36 h. Intraday precision was performed to reused the same analyte six times in the same day for target products, and interday precision was determined on six consecutive days. Intra- and interday variations of the samples were less than 0.73% and 1.75% as RSD values, respectively. Determining the repeatability of IL solution, six same extracts were successively obtained. The extraction yields of AME and HIN had good repeatability with RSD values of 1.34% and 1.72%, respectively, which suggested that the proposed method can be considered suitable for their quantification (see Table 4). The recovery for two biflavonoids was employed by adding severally target products with known content into 2.0 g powders of *S. sinensis* (n = 6). The established method had acceptable recoveries for AME and HIN in the range from 94.01% to 97.98% (RSD = 1.81%) and 97.92% to 101.16% (RSD = 1.88%), respectively. Based on the above results, the proposed analysis methods for quantification of AME and HIN in the extract were credible.

### 2.8. Mechanism of MAE-IL

Newly, MAE-IL technology has been successfully used in the extraction of flavonoids, polysaccharides, terpenes, anthraquinones and alkaloids of TCM [36] for novel industrial scale extraction systems. It can be concluded that there are two main inherent advantages existing in the MAE-IL process that achieve the high extraction performance of AME and HIN. 

First, based on the literature [37,38], imidazolium ILs possess the capacity of dissolving the cellulose in the cell walls due to their particular physicochemical properties including π-π interaction and hydrogen bonds. Hence, the use of [C_6_mim]BF_4_ as an extractant in the optimized process will certainly promote the disruption of the cell walls in *S. sinensis* and will improve the mass transfer rate of solvent permeation into the cell, thus a large number of target components will be extracted. Meanwhile, suitable IL concentration can be regarded as the primary influence factor for the extraction process. If IL concentration is too high, the extraction yields of the target constituents will constantly decrease because of the viscosity enhancement and the diffusion ability decline of IL solution.

Second, the microwave method is proposed based on the principle of electromagnetic wave, cavitation and sonoporation [39]. High-frequency electromagnetic wave generated by microwave radiation can penetrate into the extraction medium and reach the internal tissue of the cell from *S. sinensis*, and then can be quickly converted to heat energy, which will rise rapidly the temperature inside the plant cell. When the pressure inside the cell exceeds the bearing capacity of the plant issue, the plant cell wall will be ruptured. Then, AME and HIN will flow out of the cell and will be dissolved in IL solution at low temperature. Notably, Ding et al. [40] pointed out the physical phenomena generated by microwave during the extraction process, which distinctly resulted in the fragmentation and deformations on the surface of plant issues by their microscopic morphological properties of *Radix peucedani* after a period of microwave extraction [41]. In this work, MAE-IL can utilize these advantages to insure the high extraction yields of AME and HIN (as shown in Figure 11). 

## 3. Material and Methods

### 3.1. Materials

The herb was collected at Jinpenshan (Zunyi, China) on 12/07/2018, dried naturally at room temperature and smashed through the 100 mesh sieve (15–68 μm) at a super micro mill (JMS60; Shanghai Xianfeng Electrical Co., Ltd. Shanghai, China). All ionic liquids were purchased from Zhongke Kete technology industry (Lanzhou, China) (see Table 5). AME and HIN standards (>98% purity) were purchased from Chengdu Ruifensi Biotechnology Co Ltd (Chengdu, China). HPLC methanol and acetonitrile were purchased from Chengdu Kelon Chemical Reagent Factory. All other analytical grade agentias were purchased from Shuangjv Chemical Reagent Factory (Zunyi, China). 

### 3.2. MAE-ILs Extraction

In a flask, 8.0 g of *S. sinensis* powder were added with different concentration ILs which were prepared by dissolving the IL into ethanol solution [42]. Then the flask was placed in a microwave instrument (model MCR-3 from Hongsheng Co., Ltd. Min Hang, Shanghai, China) for the extraction of AME and HIN in a certain microwave power at certain temperature for some duration. After reaction, then centrifugation was performed at 4000 rpm for 10 min. And the supernatant was separated and stored at 5 °C for further analysis. The concentration of AME and HIN in the extract was quantified by high performance liquid chromatography (HPLC), as shown in Figure 12. 

### 3.3. Determination of AME and HIN Content

The HPLC analysis of AME and HIN was carried out on the basis of Chinese Pharmacopoeia (2015 edition), which was performed on a C_18_ chromatographic column (4.6 × 250 mm, 5 μm, Phenomenex, USA) at a column temperature of 25 °C. For AME and HIN, the mobile phase was composed of acetonitrile (A) and 0.1% formic acid water solution(B); the gradient elution program was as follows: 0–10 min, 40–50% A; 10–20 min, 50–60% A; 20–30 min, 60–70% A; 30–40 min, 70–80% A. The detection wavelength was set at 330 nm and the injection volume was 20 μL. It was found that AME and HIN were the major biflavonoids in the extracts at a retention time of 8.35 min and 14.59 min, respectively (see Figure 9a,b). Moreover, two biflavonoids content of *S. sinensis* extract were obtained according to the following formula:
(3)$$\mathrm{Yield}(\mathrm{mg}/g)=\frac{\mathrm{mean}\;\mathrm{mass}\;\mathrm{of}\;\mathrm{analytes}\;\mathrm{in}\;\mathrm{the}\;\mathrm{extract}(\mathrm{mg})}{\mathrm{mean}\;\mathrm{mass}\;\mathrm{of}\;S.\mathrm{sinensis}\;\mathrm{powder}(g)}\times 100$$
where mean mass of *S. sinensis* powder referred to the average mass of three samples before extraction. The mean mass of analytes in the extract was analyzed by HPLC.

### 3.4. Experimental Design

A Box–Behnken design of RSM was used to analyze and optimize the MAE-IL process of two target compounds in regard to three variables (A: [C_6_mim]BF_4_ concentration; B: extraction temperature; C: solid/liquid ratio). As shown in Table 6 and Table 7, the experiments offered different level of each factor by the factorial design and included a total of 17 runs. Each run was employed three times, and the index of evaluation were the mean values of the extraction yields. The extraction yields of AME and HIN were defined as responses to associate three independent variables, which could be expressed as the following equation:Y = β_0_ + β_1_X_1_ + β_2_X_2_ + β_3_X_3_ + β_11_X_12_ + β_22_X_22_ + β_33_X_32_ + β_12_X_1_X_2_ + β_13_X_1_X_3_ + β_23_X_2_X_3_.(4)

### 3.5. Recovery of Two Biflavonoids and IL

After extraction process, the recovery of two biflavonoids and IL from *S. sinensis* extract was investigated as following: The IL extract was diluted 10 times with deionized water. And then AME, HIN and IL were extracted using insoluble organic reagents (Vorganic reagent: Vextract = 3:1, V/V). The recovery of two biflavonoids was tested based on the HPLC method described earlier in Section 3.3. The recovery of IL was determined by HPLC-DAD, which was performed on a C_18_ chromatographic column (4.6 × 250 mm, 5 μm, Phenomenex, aberdeen, Scotland) at a column temperature of 25 °C using methanol- water (10:90, v/v) as the mobile phase. The detection wavelength was set at 210 nm and the injection volume of the sample was 10 μL. The standard curves of IL was: Y = 3287.8X + 8.0667 (r = 0.9992) (0.16–0.96 μg/μL).

### 3.6. Conventional Reference Extraction Methods

According to the extraction methods of Chen S D et al. and Eibes G et al. [43,44], ethyl alcohol was selected as the reference solvent in the MAE, soxhelt extraction (SE) and percolation extraction (PE) of AME and HIN from S. sinensis, and the extraction tests were employed in the optimized conditions for each extraction method. In short, 8.0 g of herb powders were mixed with ethanol in a 250 mL flask, and then the reaction liquids were filtrated. The suspensions were concentrated (4000 rpm, 5 °C) and dried under oven. Three different extract solutions were filtered for further HPLC analysis.

### 3.7. Statistical Analysis

Design Expert 8.0 (DE, Stat-Ease, Inc., Minneapolis, MN, USA) was adopted for designing and optimizing the process. The analysis of variance (ANOVA) data by RSM was used to analyze the experiment results and predicte values of AME and HIN, which were statistically significant at the level of *p* < 0.05. After repeated extraction tests were completed (n = 3), mean and standard deviation were calculated. 

## 4. Conclusions

In this work, IL solution combined with MAE was studied in detail on the extraction of AME and HIN from *S. sinensis* for the first time. After screening different ILs, it was found [C_6_mim] BF_4_ had significant influence on target constituents and was finally chosen as the optimal IL. The results after optimizing MAE-IL conditions by RSM indicated that the extraction yields of AME and HIN could reach 1.96 mg/g and 0.79 mg/g, respectively. In addition, compared with conventional solvent extraction, MAE-IL system showed remarkable characteristics of less solvent, higher efficiency and shorter extraction time, which confirmed that MAE-IL had the potentiality for extracting AME and HIN from *S. sinensis*. 

## Figures and Tables

**Figure 1 molecules-24-02507-f001:**
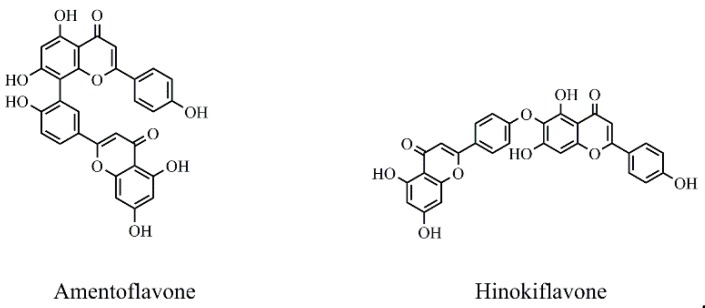
Chemical structures of two biflavonoids from *S. sinensis.*

**Figure 2 molecules-24-02507-f002:**
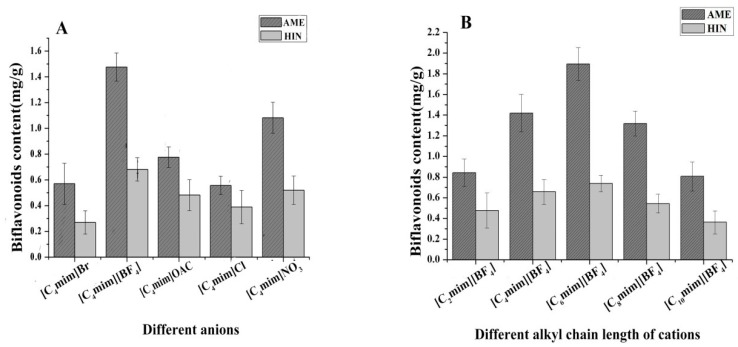
Effect of the (**A**) IL anions and (**B**) IL cations for two biflavonoids content from *S. sinensis.*

**Figure 3 molecules-24-02507-f003:**
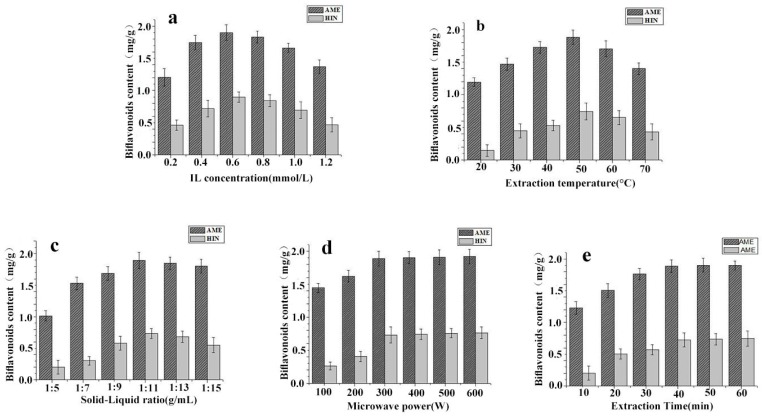
Effects of extraction parameters on two biflavanoids yields of *S. sinensis*. (**a**) IL concentration; (**b**) extraction temperature; (**c**) ratio of solid to liquid; (**d**) microwave power; (**e**) extraction time.

**Figure 4 molecules-24-02507-f004:**
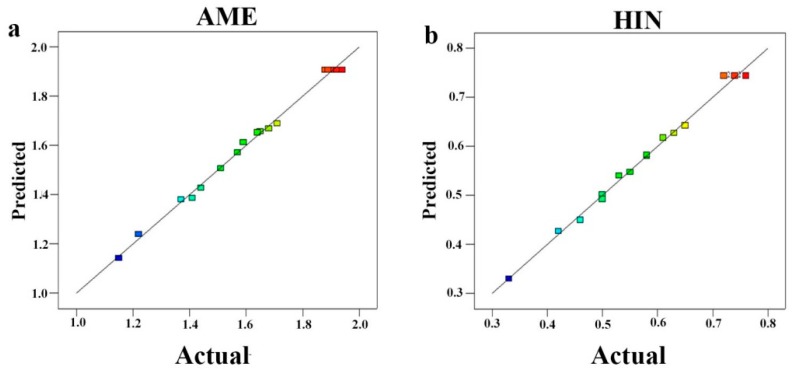
Correlation between the predicted and actual values for two biflavonoids content; (**a**) color points from blue to red show values of AME changing in the range of 1.0–2.0 mg/g; (**b**) color points from blue to red show values of HIN changing in the range of 0.3–0.8 mg/g.

**Figure 5 molecules-24-02507-f005:**
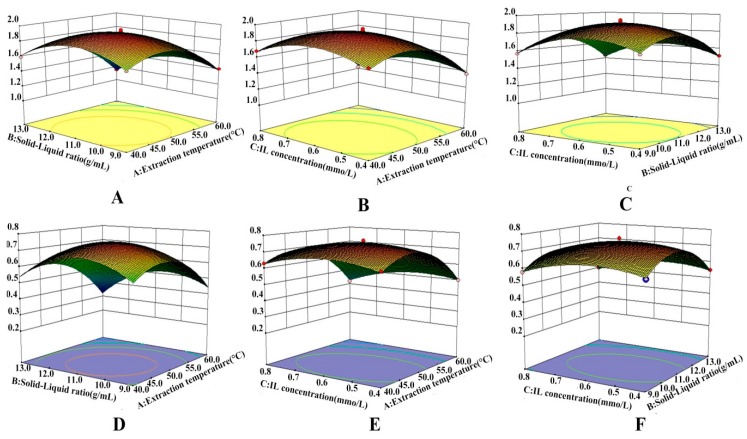
Response surface plots showing interaction effects of solid–liquid ratio and extraction temperature (**A**); extraction temperature and IL concentration (**B**); solid–liquid ratio and IL concentration (**C**) on AME yield and solid–liquid ratio and extraction temperature (**D**); extraction temperature and IL concentration (**E**); solid–liquid ratio and IL concentration (**F**) on HIN yield.

**Figure 6 molecules-24-02507-f006:**
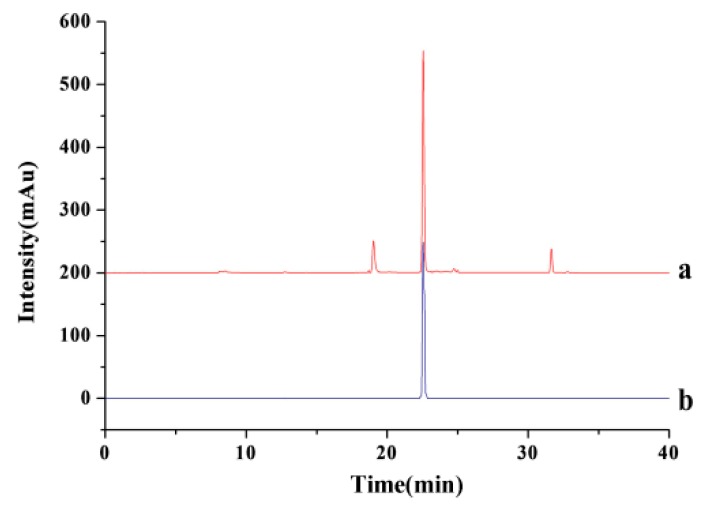
HPLC-DAD chromatograms of recovery sample from *S. sinensis* (**a**) and [C_6_mim]BF_4_ standard (**b**).

**Figure 7 molecules-24-02507-f007:**
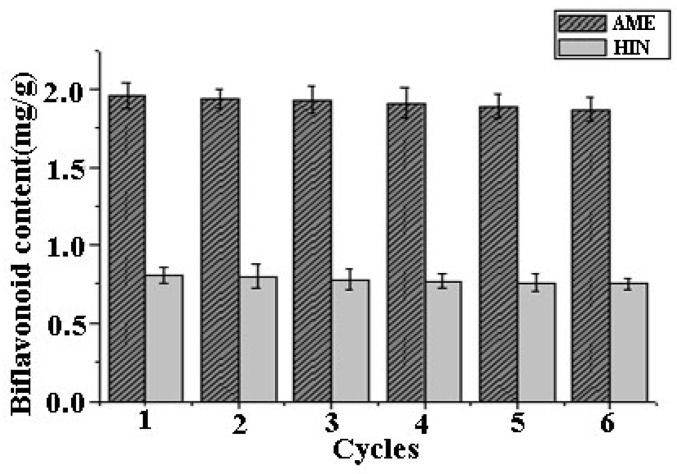
Performance of recovered [C_6_mim]BF_4_ on the yields of AME and HIN.

**Figure 8 molecules-24-02507-f008:**
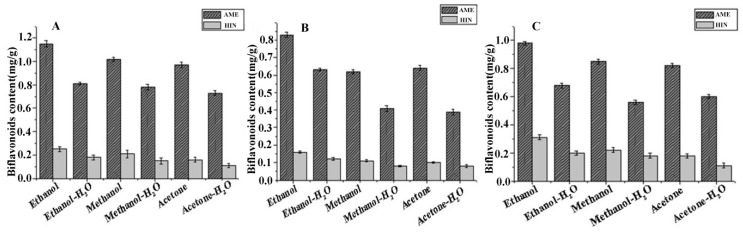
Comparison of different extractants in three conventional methods for extraction of AME and HIN from *S. sinensis.* (**A**) microwave extraction; (**B**) soxhlet extraction; (**C**) percolation extraction.

**Figure 9 molecules-24-02507-f009:**
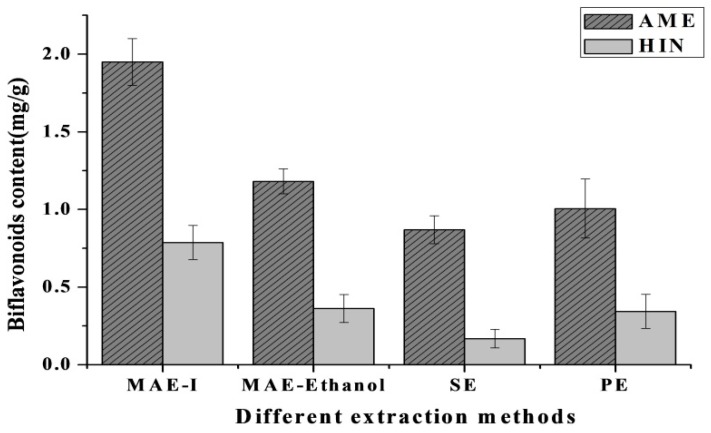
Effect of four extraction methods for biflavonoids content from *S. sinensis*.

**Figure 10 molecules-24-02507-f010:**
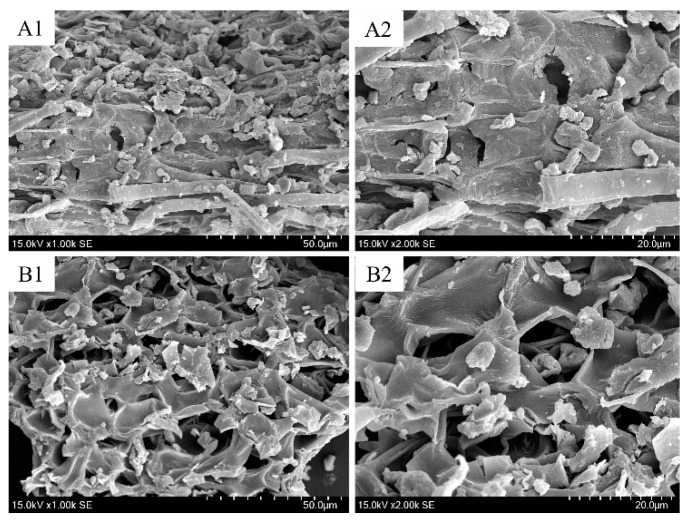
SEM graphics of *S. sinensis* before and after extraction. Raw materials (**A1**,**A2**) and treated samples by MAE-IL (**B****1**,**B2**) were observed under 1000 and 2000 magnification, respectively.

**Figure 11 molecules-24-02507-f011:**
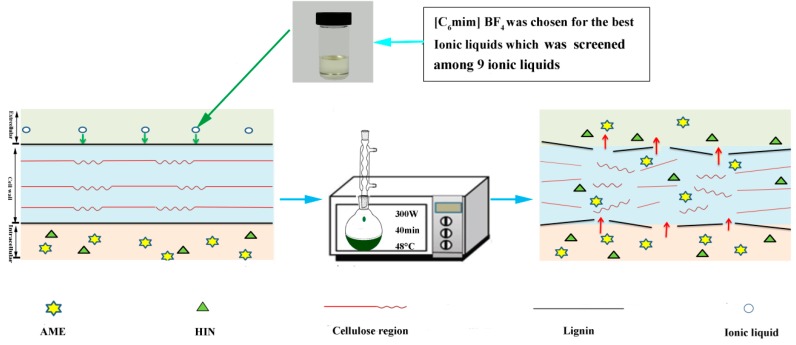
Contribution of IL and MAE to improve the extraction efficiency of AME and HIN.

**Figure 12 molecules-24-02507-f012:**
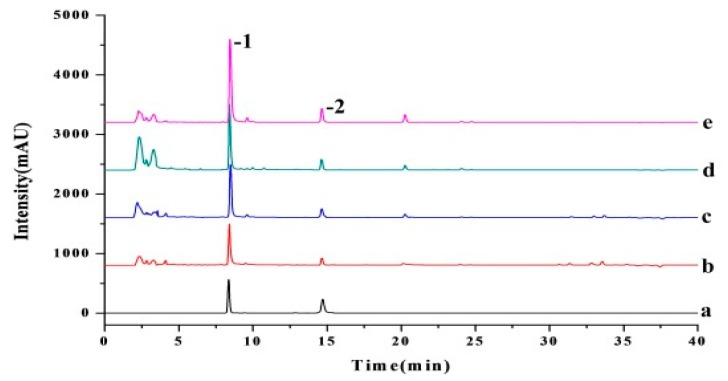
Comparison of different extraction methods for extraction of AME and HIN from *S. sinensis.* (**a**) mixed standards; (**b**) Soxhlet extraction; (**c**) Percolation extraction; (**d**) MAE-ethanol; (**e**) MAE-IL; The peaks marked with 1, 2, which were AME and HIN, respectively.

**Table 1 molecules-24-02507-t001:** ANOVA for the fitted secondary order curve for AME content.

Source	Sum of Squares	df	*F Value*	*p-Value*	*R* ^2^	*R* ^2^ *(Adj)*	*Significant*
Model	0.96	9	157.933	<0.0001	0.9951	0.9888	*significant*
X_1_X_2_	0.27	1	405.2008	<0.0001			
X_1_X_3_	0.042	1	62.23044	<0.0001			
X_2_X_3_	0.013	1	18.94292	0.0033			
X_1_^2^	0.01	1	14.79915	0.0063			
X_2_^2^	3.60 × 10^−3^	1	5.327696	0.0543			
X_3_^2^	0	1	0	1.0000			
Residual	0.27	1	394.1389	<0.0001			
Lack of Fit	0.18	1	265.7134	<0.0001			
Pure Error	0.11	1	162.5243	<0.0001			
Cor Total	4.73 × 10^−3^	7					
X_1_X_2_	2.45 × 10^−3^	3	1.432749	0.3577			*not significant*
X_1_X_3_	2.28 × 10^−3^	4					
X_2_X_3_	0.97	16					

**Table 2 molecules-24-02507-t002:** ANOVA for the fitted secondary order curve for HIN content.

Source	Sum of Squares	df	*F Value*	*p Value*	*R* ^2^	*R* ^2^ *(Adj)*	*Significant*
Model	0.258677	9	128.1486	<0.0001	0.9940	0.9862	*significant*
X_1_X_2_	0.0578	1	257.707	<0.0001			
X_1_X_3_	0.0128	1	57.07006	0.0001			
X_2_X_3_	0.00405	1	18.05732	0.0038			
X_1_^2^	0.0016	1	7.133758	0.0320			
X_2_^2^	0.0009	1	4.012739	0.0852			
X_3_^2^	1 × 10^−4^	1	0.44586	0.5257			
Residual	0.081938	1	365.3282	<0.0001			
Lack of Fit	0.070612	1	314.8287	<0.0001			
Pure Error	0.012506	1	55.76064	0.0001			
Cor Total	0.00157	7					
X_1_X_2_	0.00045	3	0.535714	0.6823			*not significant*
X_1_X_3_	0.00112	4					
X_2_X_3_	0.260247	16					

**Table 3 molecules-24-02507-t003:** Results of recovery of two biflavonoids and IL with different organic solvents (%) (n = 3).

	Ethyl Acetate	Chloroform	n-Butanol	Dichloromethane	Ether	n-Hexane
AME	92.17 ± 0.38	31.57 ± 0.29	69.63 ± 0.56	41.31 ± 0.44	81.28 ± 0.62	12.46 ± 0.37
HIN	91.53 ± 0.49	28.64 ± 0.51	63.79 ± 0.36	39.28 ± 0.41	76.17 ± 0.46	13.59 ± 0.31
IL	95.28 ± 0.37	93.36 ± 0.44	94.19 ± 0.28	94.08 ± 0.35	96.21 ± 0.32	95.89 ± 0.26

**Table 4 molecules-24-02507-t004:** Method validation for two standard compounds.

Analyte	Calibration Curve	*R* ^2^	LOD (μg/mL)	LOQ (μg/mL)	Intraday Precision (п = 6) ^a^	Interday Precision (п = 6) ^a^	Stability (п = 5) ^a^	Repeatability (n = 6) ^a^	Recovery (n = 6) ^a^
AME	y = 346.35x + 3.4667	0.9992	0.033	0.179	0.56	1.75	0.68	1.34	1.81
HIN	y = 264.05x + 6.0667	0.9996	0.046	0.278	0.73	1.39	1.21	1.72	1.88

LOD, limit of detection (S/N = 3); LOQ, limit of quantification (S/N = 10). ^a^ Intraday precision, interday precision, stability, repeatability and recovery are expressed as the RSD (%) of peak area.

**Table 5 molecules-24-02507-t005:** Chemical structures of ionic liquids.

Ionic liquids	Cations	Anions
[C_4_mim]Cl(1-butyl-3-methylimidazolium-chloride)	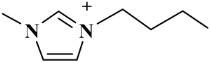	Cl^−^
[C_4_mim]NO_3_(1-butyl-3-methylimidazolium-nitrate)	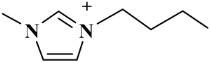	NO_3_^−^
[C_4_mim] CH_3_COO(1-butyl-3-methyli-midazolium-acetate)	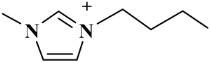	CH_3_COO^−^
[C_4_mim]Br(1-butyl-3-methylimidazolium-bromine)	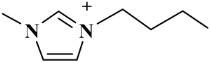	Br^−^
[C_2_mim]BF_4_(1-ethyl-3-methylimidazolium- hexafluorophosphate)	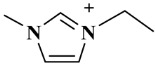	BF_4_^−^
[C_4_mim]BF_4_ (1-butyl-3-methylimidazolium-tetrafluoroborate)	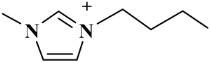	BF_4_^−^
[C_6_mim]BF_4_(1-hexyl-3-methylimidazolium-hexafluorophosphate)	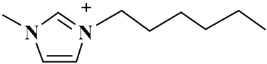	BF_4_^−^
[C_8_mim]BF_4_(1-octyl-3-methylimida-zolium-hexafluorophosphate)	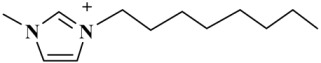	BF_4_^−^
[C_10_mim]BF_4_(1-decyl-3-methylimidazolium-hexafluorophosphate)	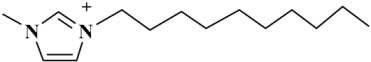	BF_4_^−^

**Table 6 molecules-24-02507-t006:** Levels and codes of factors chosen for experiment.

Level	Factors		
	Extraction Temperature (°C)	IL Concentration (mmol/L)	Solid-Liquid Ratio (g/mL)
−1	40	0.4	1:9
0	50	0.6	1:11
1	60	0.8	1:13

**Table 7 molecules-24-02507-t007:** The results of experiment design.

Run	X1 (Extraction Temperature/°C)	X2 (Solid-Liquidratio/g/mL)	X3 (IL Concentration/mmol/L)	AME (mg/g)	HIN (mg/g)
1	60	1:11	0.4	1.37	0.50
2	60	1:9	0.6	1.41	0.46
3	60	1:13	0.6	1.15	0.33
4	50	1:11	0.6	1.88	0.72
5	50	1:11	0.6	1.91	0.74
6	50	1:13	0.4	1.51	0.55
7	40	1:11	0.8	1.68	0.63
8	40	1:9	0.6	1.65	0.58
9	50	1:11	0.6	1.92	0.76
10	50	1:13	0.8	1.44	0.50
11	50	1:11	0.6	1.94	0.76
12	50	1:9	0.8	1.57	0.58
13	50	1:9	0.4	1.64	0.61
14	40	1:11	0.4	1.71	0.65
15	60	1:11	0.8	1.22	0.42
16	50	1:11	0.6	1.89	0.74
17	40	1:13	0.6	1.59	0.53

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
