# Peer review of "Ionic Liquid-Microwave-Based Extraction of Biflavonoids from Selaginella sinensis"

_molecules, 2019, doi:10.3390/molecules24132507_

Round 1
Reviewer 1 Report
The aim of this study was to develop a convenient, effective and rapid microwave assisted extraction-ionic liquids (MAE-IL) method for extraction of amentoflavone (AME) and hinokiflavone (HIN) from Selaginella sinensis. Nine different ILs was investigated and [C6mim]BF4 was found to have a high selectivity and efficiency. Some important extraction conditions including ILs concentration, extraction temperature, solid/liquid ratio, microwave power and extraction time were studied and the extraction parameters were optimized by response surface method (RSM). The present optimized MAE-IL method is better than those of MAE, soxhelt extraction (SE) and percolation extraction (PE) methods for AME and HIN from S. sinensis. There are some concerns (typos, inconsistency, and others) as listed in the following:
*P1/14: the keyword HPLC was not appeared in the Abstract.
P1/14: Amentoflavone->amentoflavone
P1/14: cyclo-oxygen-ase-2-> cyclooxygenase-2
P1/14: podocarpu- sflavone-> podocarpusflavone
P1/14: respiratory syncytial virus pneumonia (RSV)-> respiratory syncytial virus (RSV) pneumonia
P1/14: Herpes Simplex Virus-> herpes simplex virus
*P1/14: decrease the mitochondrial membrane?
*P2/14: structurally asymmetric anions and anions[12].?
P3/14: liquid/solid ratio vs. P4/14: solid-liquid ratio
P3/14: Effect of ILs concentration-> Effect of ILs Concentration
*P3/14: in the range of 20 to 50 °C, the extraction efficiency of AME and HIN would continue to add with the increase of solid-liquid ratio? as a result of the solubility and permeability increasing of [C6mim] BF4 solution.
P3/14: could generated-> could generate
P5/14: ultrasound power-> microwave power
*P5/14: 3.3. Analysis of Response Surfaces-> 2.3
P5/14: Content;? (a)
*P6/14: 1:11 g/mL ratio of liquid to raw?
P7/14: 2.4. Recovery of two biflavonoids and IL
P7/14: HPLC analysis datas-> HPLC analysis data
P7/14: SE and PE-> full name first
*P7/14: 2.5. Comparison of Five? Extraction Methods
P7/14: content MAE?
P8/14: 2.6. Feedstock analysis before and after extraction
P8/14: aplied to observe the characterize
P8/14: (Figure 9A, A1)
P8/14: by SME
*P8/14: LOQs (S/N= 10) and LODs (S/N=3) for AME and HIN of the extracts were less than 0.278 mg/mL and 0.046 μg/mL, respectively.
P11/14: Table 2: Extract Temperature
R6: 335-40
R9: Inhibitory Effect of Selaginellins….: capital letter
*R16: Ionic Liquid-Ultrasound-Based Extraction…: capital letter; 1420-3049?
Author Response
Dear Reviewer:
We do extremely appreciate your suggestions. Based on them, we have made careful investigation, revision and improvement for the original manuscript. Main changes in the revised manuscript have been marked as highlighted text. We hope the revised manuscript will meet your requirement. The following content is our point-by-point responses to your comments and questions.
Looking forward to hearing from you soon, thank you very much!
Yours Sincerely
Authors
Response to reviewer
1. *P1/14: the keyword HPLC was not appeared in the Abstract.
Response: Thank you for your advice. We have added the description about HPLC in the Abstract.
2. P1/14: Amentoflavone->amentoflavone
Response: We have revised the error in the Introduction.
3. P1/14: cyclo-oxygen-ase-2-> cyclooxygenase-2
Response: Thank you very much, We have revised the spelling error.
4. P1/14: podocarpu- sflavone-> podocarpusflavone
Response: We have revised the spelling mistake.
5. P1/14: respiratory syncytial virus pneumonia (RSV)-> respiratory syncytial virus (RSV) pneumonia
Response: Thank you for this direction, We have carefully revised the error in the Introduction.
6. P1/14: Herpes Simplex Virus-> herpes simplex virus
Response: Sorry, We have corrected the spelling mistake.
7. *P1/14: decrease the mitochondrial membrane?
Response: Thank you very much. We have changed “mitochondrial membrane” to “mitochondrial membrane potential”.
8. *P2/14: structurally asymmetric anions and anions[12].?
Response: Many thanks for your advice. This was our mistake. We have changed “anions and anions” to “anions and cations”.
9. P3/14: liquid/solid ratio vs. P4/14: solid-liquid ratio
Response: We have carefully corrected the error throughout the manuscript.
10. P3/14: Effect of ILs concentration-> Effect of ILs Concentration
Response: We have revised the spelling mistake.
11. *P3/14: in the range of 20 to 50 °C, the extraction efficiency of AME and HIN would continue to add with the increase of solid-liquid ratio? as a result of the solubility and permeability increasing of [C6mim] BF4 solution.
Response: Thanks for your comment! This was our mistake. We have changed the sentence to “the extraction efficiency of AME and HIN would continue to add with the increase of temperature”
12. P3/14: could generated-> could generate
Response: We have corrected the grammar error.
13. P5/14: ultrasound power-> microwave power
Response: We have revised the spelling mistake in the manuscript.
14. *P5/14: 3.3. Analysis of Response Surfaces-> 2.3
Response: Thank you very much. We have carefully checked and revised the numbering of sections and subsections throughout the manuscript.
15. P5/14: Content;? (a
Response: Sorry, we have deleted the word.
16. *P6/14: 1:11 g/mL ratio of liquid to raw?
Response: Many thanks for your advice. We have changed “ratio of liquid to raw” to “ratio of solid to liquid”.
17. P7/14: 2.4. Recovery of two biflavonoids and IL
Response: We have changed the caption to “Recovery of Two Biflavonoids and IL” in section 2.4.
18. P7/14: HPLC analysis datas-> HPLC analysis data
Response: We have modified the grammar error.
19. P7/14: SE and PE-> full name first
Response: We have added the full name of the extraction methods in the manuscript.
20. *P7/14: 2.5. Comparison of Five? Extraction Methods
Response: Thank you for this direction. We have changed “Comparison of Five” to “Comparison of Four”.
21. P7/14: content MAE?
Response: Sorry, we have deleted the word.
22. P8/14: 2.6. Feedstock analysis before and after extraction
Response: We have changed the caption to “Feedstock Analysis Before and After Extraction” in section 2.6.
23. P8/14: aplied to observe the characterize?
Response: Thank you very much. We have revised the spelling mistake in the sentence.
24. P8/14: (Figure 9A, A1)
Response: We have revised the caption sequence in the figure.
25. P8/14: by SME?
Response: We have changed “SME” to “SEM”.
26. *P8/14: LOQs (S/N= 10) and LODs (S/N=3) for AME and HIN of the extracts were less than 0.278 mg/mL and 0.046 μg/mL, respectively.
Response: Thank you very much for your guidance. We have changed “mg” to “μg” in the sentence.
27. P11/14: Table 2: Extract Temperature
Response: We have revised the error in Table 6 and Table 7.
28. R6: 335-40?
Response: We have rechecked all cited references and replaced some inappropriate references, for example reference 6.
29.R9: Inhibitory Effect of Selaginellins….: capital letter.
Response: We have revised the format errors in reference 9.
30.*R16: Ionic Liquid-Ultrasound-Based Extraction…: capital letter; 1420-3049?
Response: Thank you for this direction. We have carefully revised the format mistakes and the error page number in reference 16.
Reviewer 2 Report
This is an interesting manuscript concerning development of the method of ionic liquid-microwave-based extraction of biflavonoids, amentoflavone and hinokiflavone from traditional Chinese herb Selaginella sinensis. Single factor experiment and response surface methodology were used for method optimization. The developed method was compared to conventional extraction methods and was proposed as quick, efficient and environment friendly extract method.
The manuscript is well organized and I believe it is acceptable for publication in Molecules after minor corrections listed below:
- Cited literature throughout Introduction section should be revised, as in few cases it seems to be inappropriate, for example reference 12 concerning ionic liquids.
- “traditional extraction methods” replaced with “conventional extraction methods” throughout the text.
- when discussing the effect of ILs concentration in subsection 2.2.1. maybe it should be better to say “the extraction yields of AME and HIN in the range of 0-0.6 mol/L was linearly proportional to the concentration of (C6mim)BF4 “ instead of “concentration of (C6mim)BF4 was linearly proportional to the extraction yields of AME and HIN in the range of 0-0.6 mol/L “
- in Section 3.3, the proposed formula for AME or HIN content (mg/kg) should be extended, with clearly denoted how the AME or HIN amount is obtained.
- Section 3.6. Conventional Reference Extraction Methods should be improved, for example instead of reaction solution ethanolic solution.
- English language and style should be improved.
- Numbering of sections and subsections sould be correcteed
After these modifications, I recommend the publication of the article.
Author Response
Dear Reviewer:
We do extremely appreciate your suggestions. Based on them, we have made careful investigation, revision and improvement for the original manuscript. Main changes in the revised manuscript have been marked as highlighted text. We hope the revised manuscript will meet your requirement. The following content is our point-by-point responses to your comments and questions.
Looking forward to hearing from you soon, thank you very much!
Yours Sincerely
Authors
Response to reviewer
1. Cited literature throughout Introduction section should be revised, as in few cases it seems to be inappropriate, for example reference 12 concerning ionic liquids.
Response: Thank you for your advice. We have rechecked all cited references and replaced some inappropriate references, for example reference 4,6,7,8,10 and 12.
2. “traditional extraction methods” replaced with “conventional extraction methods” throughout the text.
Response:We have changed “traditional extraction methods” to “conventional extraction methods” throughout the manuscript.
3. When discussing the effect of ILs concentration in subsection 2.2.1. maybe it should be better to say “the extraction yields of AME and HIN in the range of 0-0.6 mol/L was linearly proportional to the concentration of (C6mim)BF4 “ instead of “concentration of (C6mim)BF4 was linearly proportional to the extraction yields of AME and HIN in the range of 0-0.6 mol/L
Response: Many thanks for your advice. We have change the sentence to “the extraction yields of AME and HIN was linearly proportional to the concentration of (C6mim)BF4 in the range of 0~0.6 mmol/L” in subsection 2.2.1.
4. In section 3.3, the proposed formula for AME or HIN content (mg/kg) should be extended, with clearly denoted how the AME or HIN amount is obtained.
Response: Thanks for the very meaningful suggestion! We have found the problem and revised the calculation formula for AME and HIN content in detail in section 3.3.
5. Section 3.6. Conventional Reference Extraction Methods should be improved, for example instead of reaction solution ethanolic solution
Response: Thanks for your comment! We investigated different extractants (ethanol, ethanol-H2O (1:1,v/v), methanol, methanol-H2O (1:1,v/v), acetone-H2O (1:1,v/v)) on the extraction performance of AME and HIN with the conventional approaches, i.e. MAE, SE and PE. Among them, we found ethanol could obtain high yields of two biflavonoids with the three methods and was suitable for further evaluated for the comparison between MAE-IL and the conventional methods. The study have been written in the manuscript.
6. English language and style should be improved.
Response: Thanks for the very meaningful suggestion! We have rechecked and revised English language and style in detail in the manuscript.
7. Numbering of sections and subsections sould be correcteed
Response: Thank you for this direction. We have carefully rechecked and corrected the numbering of sections and subsections throughout the manuscript.